# Epidemiological Analysis of Suicidal Behaviour in Spain from 2017 to 2022 and Comparative Perspectives with Japan: A Retrospective Observational Study

**DOI:** 10.3390/healthcare13050451

**Published:** 2025-02-20

**Authors:** Noelia L. Martínez-Rives, Hellen W. Babu, Yasuhiro Kotera, Pilar Martin, Rory D. Colman, Stuart Gilmour

**Affiliations:** 1Department of Psychiatry and Social Psychology, University of Murcia, 30100 Murcia, Spain; noelialucia.martinezr@um.es; 2Graduate School of Public Health, St. Luke’s International University, Tokyo 104-0045, Japan; hellenbabu@gmail.com (H.W.B.); sgilmour@slcn.ac.jp (S.G.); 3Faculty of Medicine and Health Sciences, University of Nottingham, Nottingham NG7 2TU, UK; yasuhiro.kotera@nottingham.ac.uk; 4Center for Infectious Disease Education and Research, Osaka University, Osaka 565-0871, Japan; 5College of Health, Psychology and Social Care, University of Derby, Derby DE22 1GB, UK; rory.d.colman@gmail.com

**Keywords:** suicide, epidemiological analysis, COVID-19 pandemic, Spain, Japan

## Abstract

Background: Suicide is a significant public health issue globally. The patterns and disparities in suicide rates may have changed with the onset of the COVID-19 pandemic. Using epidemiological comparative studies, we can identify how suicide incidence is influenced by risk factors such as personal pressures and social support. This study analyses Spanish suicide data during the period from 2017 to 2022 and compares them with previously analysed data from Japan, with the aim of exploring the variability in suicide distribution in both countries. Methods: We used a retrospective observational design with national-level suicide data from Spain to estimate trends in suicide mortality over this period and compared these data with previously analysed data from Japan. Results: In Spain, no significant changes in suicide rates were found before and after the pandemic period, but notable monthly variations occurred at the pandemic’s onset in the youngest age group, highlighting the increase use of poisoning among women until 2020. In Japan, a notable variation was found following the pandemic. Specifically, older age groups and younger women were at higher suicide risk, while the suicide risk transitioned from younger to older ages between 2020 and 2021 among men. These changes are likely attributed to exacerbated socio-economic factors. A similar trend was observed in both countries based on sex with a different influence noted for women. Conclusions: This study offers detailed insight into the patterns and circumstances of suicide in these countries, offering a basis for future interventions and preventive measures. These comparisons emphasize the critical nature of suicide and underline the necessity for enhanced prevention efforts.

## 1. Introduction

### 1.1. Suicide Worldwide

Figures show that more than 720,000 people die due to suicide worldwide annually [1], with 759,028 cases reported in 2019, reaching global age-standardized rates of 9 cases per 100,000 people in both sexes. In addition, men have a rate more than double that of women (12.6 in males vs. 5.4 in females) [2]. Globally, 73% of all completed suicides take place in low- and middle-income countries, showing differences with respect to factors like age, sex, and suicide methods, which are affected by social, cultural, economic and environmental factors [1].

According to data from the World Health Organization (WHO), there are significant differences in suicide rates globally, and they vary across regions and countries. In 2012, some regions, like South-East Asia, had a smaller percentage of the world’s population but a higher proportion of suicides. Also, when comparing age-standardized suicide rates, a great deal of variability was observed in populations with more than 300,000 people, where suicide rates were much higher than in others. These differences in suicide rates between countries have been consistent over time [3].

### 1.2. Suicide in Spain and Japan

Specifically in Spain, suicide rates are between 5.0 and 9.9 [4], with 6.6 reported for both sexes in 2019 (7.9 in males and 2.8 in females) [2]. If we compare these data with a country with the highest rates of suicide of ≥15, like Japan (18.1 in both sexes in 2019, 17.5 in males and 6.9 in females) [2], which is the first on the list when WHO began to record data on this problem [4], Spanish data may not appear significantly high. However, the upward trend observed in recent years raises concerns about future developments.

Regarding suicide data collected from Spain before the year 2017, some authors stressed the idea that the National Institute of Statistics (INE) records, at least until that date, registered fewer suicides than real cases. Improvements in registration could partly explain the increase in cases after that date [5]. The study in Japan by Matsubayashi and Ueda [6] also included the period prior to 2017. Contrary to Spanish records, their results revealed no correlation between deaths from unknown causes and suicide rates or socioeconomic indicators, suggesting that suicide statistics are accurate in Japan.

Previous studies of suicide statistics in Spain between 1981 and 2008 showed a decrease since 1990, except for some peaks that occurred in times of crisis [7]. Analysing differences based on sex, an always repeated phenomenon was revealed, namely, the predominance of male suicide. However, an increase among women aged 35–49 years was observed [7]. As we have seen in studies that covered periods after this date (2008–2017) [8], similar sex trends were observed in the Spanish region of Andalusia with the highest total number of suicides, but not rate, in those years. Regarding region, rural populations and areas with high levels of unemployment stood out. In contrast, previous research from 1910 to 2011 [9], which considered socioeconomic variables, contradicted hypotheses relating suicide to urbanisation processes and poverty.

In Europe, hanging seems to be the most common suicide method [4]. A study conducted in Spain [10] highlighted the predominance of non-toxic violent methods, although poisoning stood out in women. Specifically in Spain [4], falls stood out in addition to hanging, where women were noted in 37% of cases. In Japan, the method used for female suicide did not differ from the method mostly used in men, namely, hanging, at 60% [4]. Although based on older Japanese data, up to 2016, the most widely used method in younger age groups (19 to 20-year-olds) was hanging, increasing since 1998 [11]. Suicide rates and the methods used can vary significantly between men and women. Although a more homogeneous trend is found in terms of sex, more completed suicides are found in males compared with females in general. Japan is one of the countries with the highest rates of suicides in women. Methods vary more depending on the region, but mostly hanging, self-poisoning, and firearms stand out [4].

Suicide incidence can show seasonal or monthly variations. For example, some studies have found peaks in certain months of the year, which may be related to climate [12]. In Spain, a unimodal seasonal pattern was observed prior to 2017, with a relatively small portion of the weekly variation explained by seasonal components. The relative risks (RRs) for suicide seasonality prior to 2017 in both Spain and Japan ranged between 1.2 and 1.3, indicating moderate seasonal effects. The seasonal pattern differed by age groups in both countries, with a larger amplitude noted in older adults compared to younger individuals (*p* < 0.05). Seasonal patterns also exhibited significant sex differences in Japan (*p* < 0.05), with male suicides peaking approximately three weeks earlier than female suicides [13]. Comparing these monthly patterns between Spain and Japan can offer insights into how temporal variations affect different populations.

### 1.3. Impact of the COVID-19 Pandemic

The COVID-19 pandemic was associated with an increase in suicide rates in Spain, with an increase of 250% in 2020, in contrast to earlier years. Specifically, 3941 people died by suicide, 7.4% more than in 2019 [14]. Between 2017 and 2022, a study focusing on cases treated by hospital emergency services in Catalonia reported an increase in suicidal ideation among younger and lower-risk profiles during the COVID-19 pandemic [15]. Other regions of Spain were also studied in detail in healthcare centres, discovering an important increase in suicidal behaviour in children and adolescents between 2018 and 2021 in the province of Alicante [16]. In Salamanca, data were analysed from minors under 18 years of age who needed help from health services due to suicidal behaviour from 2019 to 2022, obtaining higher numbers in 2022 compared to previous years, and highlighting an increase in suicidal ideation especially in women. In addition, the annual incidence in men was more than double [17].

Spanish national studies examined the impact of the COVID-19 pandemic on suicide from 2016 to 2020, showing an increase at the beginning of the pandemic outbreak and in older adults [18]. This pattern highlights the vulnerability of elderly individuals during times of prolonged social isolation and health crises. A longitudinal analysis of suicide data between 2000 and 2021 in Spain indicated a significant increase in suicide mortality since the onset of the pandemic, particularly among middle-aged adults, residents of large urban areas, and single individuals [19].

Suicide statistics in Japan also reported an increase in suicide cases in the year of the pandemic, 2020, showing an important increase in working-age female suicides as a high-risk group for this period [20]. In addition, changes in the reasons for suicide in Japan depending on sex were observed during the COVID-19 pandemic [21]. The report from the Ministry of Health, Labour and Welfare in Japan in the year 2020 showed higher figures in the age range of 20–29 years old [22]. The motives of the trends in suicidal mortality were analysed, finding that the suicide rates of both sexes between 20 and 29 years increased in the late 2010s, attributed to depression. For the causes, they mention problems related to economics, employment, family, and romance [23].

### 1.4. The Role of Comparative Studies in Understanding the Epidemiology of Suicide

Epidemiological comparative studies have been useful to understand variations in the prevalence of diseases and their risk factors in different populations and time periods, like the study from the GBD 2019 Ageing Collaborators [24] about the burden of diseases and injuries in elderly population. Such studies also identify patterns, trends, and disparities in health issues, facilitating an understanding of how sociocultural factors influence problematic behaviours that are studied worldwide, like suicide. Comparing data across countries can help raise public awareness of the severity of the suicide problem and the need for stronger prevention efforts.

By comparing detailed data, we can observe increases or decreases in the use of certain methods or identify which ones are more used in certain age groups or sexes. In that way, we can identify which prevention strategies have been effective in one country and consider adapting them in the other. For example, Japan has developed specific programs to reduce suicide among the elderly [25]. Knowing the most vulnerable groups and the most common methods motivates more specific and effective interventions to be designed, such as awareness campaigns at key points for target population groups or restricting access to common methods of suicide.

### 1.5. Objective and Rationale of the Study

For these reasons, the main objective of this study is to analyse suicide rate patterns across different age groups in Spain during the period between 2017 and 2022, considering variations in methods, timing, and sex, and to compare the findings with those data previously analysed from Japan for the same period. This research seeks to identify specific patterns and variations that may influence the incidence and characteristics of suicide in both countries. By understanding these differences, it is hoped to contribute to the development of more effective prevention strategies adapted to the particularities of each sociocultural context.

The period from 2017 to 2022 was selected as an available interval that provides a significant amount of current data that allows identifying consistent trends and patterns over time, as well as the impact of the COVID-19 pandemic, which added significant social, economic, and cultural changes. The pandemic also affected mental health, being more relevant and applicable to current prevention strategies, especially for its psychological and social impact on children and young people [26]. The period of impact of the COVID-19 pandemic ranges from mid-March 2020, since on March 11 the World Health Organization (WHO) officially declared it a global pandemic, until May 2023 [27].

This comparison also offers a perspective of the state of these countries, Spain and Japan, with respect to this issue and a healthy competition to achieve a population as protected and prepared as possible against this international problem.

### 1.6. Research Questions

(a)What are the main differences in suicide rates across different age groups in Japan and Spain during the period between 2017 and 2022?(b)What are the differences in suicide rates between sexes in these countries during the same period?(c)Are there significant differences in the methods used and time of year (monthly patterns) of suicide?(d)Did the COVID-19 pandemic affect suicide mortality rates differently in both countries?

## 2. Materials and Methods

### 2.1. Study Design and Setting

This study uses a retrospective observational design using epidemiological data from Spain and previously analysed data from Japan. The setting includes national-level suicide statistics from 2017 to 2022, a period chosen to observe trends before, during, and after the COVID-19 pandemic. This design allows for identifying and comparing patterns in suicide rates, methods, and age- and sex-based differences between two culturally distinct countries [28]. It was conducted following the STROBE (Strengthening the Reporting of Observational Studies in Epidemiology) [29] guidelines to ensure the consistency and quality of the results obtained, and whose checklist can be found in Appendix A. All criteria for reporting observational studies were adopted and applied at each stage of the study.

Approval from a clinical research ethics committee was not required because the study is based on publicly available, anonymized aggregate data that do not involve human participants or personal identifying information.

### 2.2. Variables and Data Sources

The variables used in this study were total of suicide death rates, sex (male and female), age, month of the death, and means/methods of suicide.

In our analysis, using the International Classification of Disease (ICD-10) classification we grouped the causes of death as follows: X60–X70 as “poisoning”, X70–X71 as “hanging and drowning”, X72–X73 as “guns”, and X80–X82 as “jumping and crushing”. In addition, X75–X79 and X83–X84 were grouped as “others” [30]. Age groups were classified as follows: 0–14 years, 15–29 years, 30–39 years, 40–84 years in 5-year intervals, and 85 years and older. We calculated directly standardized mortality rates to show 5-year trends in suicide mortality using the 2019 population [31]. We also created a variable indicating suicide deaths happening before and after the pandemic to estimate the effect of COVID-19 on suicide mortality (step change). Annual suicide death data were used to estimate the trend in suicide mortality based on method over 5 years, while monthly suicide data were used to estimate time trend and pandemic effect separately based on sex.

This study used statistics from the INE (Instituto Nacional de Estadística) [32] in Spain. Microdata on cause of death statistics from 2017 to 2022 were obtained from the same webpages, including demographic characteristics of the deceased (age and sex), month in which the death occurred, and detailed methods of suicide. Japanese data used for the comparison were compiled from published studies and national reports that met established criteria for relevance and methodological rigor for the same period (2017–2022).

### 2.3. Statistical Analysis

We used Poisson regression to estimate the trends in suicide mortality based on sex, age group, method, time, and effect of COVID-19. We included an interaction term between time and the pandemic variable to estimate changes in the trend of suicide mortality due to the COVID-19 pandemic. We also included an interaction between age group and the pandemic to investigate whether the impact of the pandemic period on incidence varied by age group and to investigate the pandemic’s effects on specific age groups relative to their pre-pandemic risk. We used the log of population as an offset in our model. We used Stata MP version 18 (Stata Corp LLC, College Station, TX, USA) for all our analyses.

To adjust the suicide rates for comparisons between different age groups and for time trends, we used directly standardized rates (DSR). In the multivariate analysis for both sexes, incidence rate ratios (IRRs) with confidence intervals (CI = 95%) and *p*-values were calculated for different predictor variables of suicide mortality, indicating the relative change in the incidence rate and statistical significance, respectively. Standard errors were included to show the precision of the IRR results.

### 2.4. Risk of Bias

As this is a retrospective observational study, information bias is limited since the data analysed were previously recorded before the study was conducted and are not subject to researcher influences. Although references to data from Japan are included, it is clarified that this is not a formal comparative study. Comparative perspectives serve only as a context to enrich the discussion, avoiding undue causal inferences.

To reduce selection bias, the Spanish data used in this study came from the INE, which is an official and recognized source for mortality data, including suicide statistics. The inclusion of all cases registered at the national level were assured. Dividing the data by sex and age groups allows for differentiated analyses that minimize confounding effects related to these demographic variables, ensuring a more accurate interpretation of trends.

## 3. Results

### 3.1. Sample Description

During the period 2017–2022, 23,060 suicides were recorded in Spain at the national level, including all ages from those under 15 years of age to people aged 95 years or older. The distribution by sex shows a significantly higher prevalence in men (73% of the total). The breakdown by year was as follows [32]:-Year 2017: 3679 total cases (2718 men, 961 women).-Year 2018: 3539 total cases (2619 men, 920 women).-Year 2019: 3671 total cases (2771 men, 900 women).-Year 2020: 3941 total cases (2930 men, 1011 women).-Year 2021: 4003 total cases (2982 men, 1021 women).-Year 2022: 4227 total cases (3126 men, 1101 women).

### 3.2. General Suicide Rates by Sex, Methods, and Monthly Trends

Figure 1 shows that sex-specific suicide rates had a marked disparity over the 5-year period. Males maintained elevated rates of approximately 12 deaths per 100,000 population, while female rates remained at 4 deaths per 100,000 population. Both crude and standardized rates demonstrated stability, with a marginal increase following the onset of the COVID-19 pandemic in 2020.

We conducted analyses to examine trends in suicide mortality by method, as shown in Figure 2. For females, the most common method both before and during the pandemic was jumping and crashing, with rates rising to approximately 1.79 per 100,000 by 2022 from 1.65 per 100,000 in 2017, with a non-linear trend. A marginal upward trend was observed in suicide by poison with rates moving from 0.79 to 1.05 per 100,000 between 2017 and 2020. In comparison, males experienced higher overall suicide rates, with hanging and drowning being the most common method, increasing from about 6.5 per 100,000 in 2017 to 7.2 per 100,000 by 2022. Jumping and crashing among males rose from 2.7 to 3.18 per 100,000 over the same period. Apart from a modest increase in hanging and drowning after 2020, no substantial changes in suicide methods were observed among males.

Figure 3 shows the monthly trends in crude and directly standardized suicide rates by sex, with Point 0 marking the start of the COVID-19 pandemic in March 2020 to highlight any changes associated with its onset. There was no noticeable change in trend or step change in suicide mortality for either sex following the start of the pandemic. Male mortality rates remained consistently higher than female rates throughout the study period, with male rates fluctuating between 3.5 and 4.0 per 100,000 person-years before the pandemic. For females, both CDR and DSR remained below 1.0 per 100,000 person-years before the pandemic, with stable patterns continuing over time. The close alignment between CDR and DSR for both sexes indicates that the age structure had minimal influence on the observed trends.

### 3.3. Multivariate Analysis of Suicide Mortality

In our multivariate analysis for males, shown in Table 1, the baseline time trend (IRR = 1.0, 95% CI: 0.998–1.002) and the pandemic period (IRR = 0.1, 95% CI: 0.88–1.11) did not show significant independent associations with suicide mortality. Similarly, the interaction between time and the pandemic (IRR = 1.002, 95% CI: 0.999–1.005) was not statistically significant. The *p*-value in Table 1 indicates whether there is a significant association with suicide mortality in men. Regarding time (monthly) and the pandemic, which both had values greater than 0.05, we cannot conclude that these factors have a significant impact on the suicide mortality, while age has a significant influence on it, with risk increasing as age increases. For interactions between age groups and the pandemic, *p*-values vary. For example, in the groups under 30 years old and 60 years old, there is a significant interaction in terms of suicide mortality. However, for the rest of the age groups, *p*-values are not statistically significant, indicating that the pandemic does not seem to influence suicide mortality in them.

In the analysis for females shown in Table 2, the baseline time trend (IRR = 0.998, 95% CI: 0.995–1.002) did not demonstrate a statistically significant association with suicide mortality, as a *p*-value greater than 0.05 would indicate. Similarly, the pandemic period (IRR = 1.18, 95% CI: 0.97–1.45) was associated with a non-significant increase in suicide risk. However, the interaction between time and the pandemic period was statistically significant (IRR = 1.0057, 95% CI: 1.0008–1.0107, *p* = 0.024) with *p*-values less than 0.05, suggesting a 0.5% positive change in suicide trend due to the pandemic’s effect.

Regarding age groups, *p*-values of 0 in all age categories indicate a significant influence on suicide mortality rates, with the risk increasing as age does, like that noted in men. Relating the age variable to the pandemic, *p*-values less than 0.05 in the age groups of 40 and 75 years old indicate significant associations between them, but non-significant interactions were found between other age groups.

### 3.4. Age-Specific Suicide Risk

The analysis shows significant differences in the risk of suicide using the 15–29-year-old group as the reference. The relative risk is expressed by the IRRs represented in Figure 4 through the blue and yellow spikes, which indicates how many times higher (or lower) the probability of suicide is compared to this reference group.

Age-specific patterns revealed substantial heterogeneity in suicide risk compared to the 15–29-year-old group in both sexes. For the males, the highest baseline risk was observed in the 90–94 age group (IRR = 8.26, 95% CI: 7.02–9.7), while the lowest was observed in the 30–39 age group (IRR = 1.43, *p* < 0.001, 95% CI: 1.2–1.7). Risk generally increased across age bands until age 90, while lower risk was observed in the 0–14 age group. In females, a similar but less pronounced trend was observed. The highest risk was observed in the 55–59 age group (IRR = 3.2276, 95% CI: 2.727–3.8201), followed by the 50–54 age group (IRR = 2.998, 95% CI: 2.535–3.5456). The risk of suicide does not increase as much with age as noted in men, but there is still a notable increase in middle age. The risk of suicide relative to 15–29 years olds before the pandemic are shown in Figure 4, with a red dotted line indicating a rate ratio of 1 as a cut-off point to indicate a significance threshold.

### 3.5. Impact of the COVID-19 Pandemic

To further explore changes in suicide deaths by age group during the pandemic, we conducted linear combinations, and the resulting incidence rate ratios are illustrated in the forest plots shown in Figure 5 by showing relative risk points through the horizontal spikes and confidence intervals for each age group after the pandemic relative to their pre-pandemic risk. The grey colour shows the precise effect size, while the peak length shows the upper and lower CI of the suicide mortality rate.

In males, a higher risk of suicide during the pandemic relative to pre-pandemic was observed in 0–15 year olds (age 0) and 60–64 years olds (age 60) with 118% and 25% higher risks, respectively. No significant changes in suicide risk were observed in other age groups because of the pandemic. In females ages 45–49 years, a 23% increase in risk was observed during the pandemic compared to pre-pandemic. A non-significant effect of the pandemic on suicide risk was observed in all other age groups.

## 4. Discussion

### 4.1. Spanish Context

The results found offer a comprehensive overview of the evolution of suicide mortality in Spain during the period 2017–2022, with a specific focus on differences based on sex, methods used, and frequency or monthly variation trends. The predominance of suicides among men is in line with previous research that explains that the impact of the suicide method on the body usually coincides with its lethality in most cases, suggesting that women are less likely than men to use facially disfiguring suicide methods [33,34]. However, even people with a stronger intent to die tend to choose higher lethality methods, such as firearms or hanging. In some studies, it has been found that in men other factors also contributed to their higher suicide rates, like male socialization, social insolation that reduces the chances of being rescued, impulsive personality traits, or alcohol consumption [35].

As for age distribution, men over 90 years of age presented the highest risk of suicide, followed by a progressive increase with age. For women, the risk peaked between 55 and 59 years of age, concluding that the relationship between suicide risk and increasing age is more pronounced in men during the years 2017–2022. In an age period cohort analysis of suicide mortality in Spain from 1984 to 2018, this increase in suicide rates with age was a trend in both sexes, being even more stable in women [36].

The most common methods also varied by sex. Men preferred methods such as hanging and drowning, while women showed an increasing preference for jumping and collision. This contrasts with findings from previous years that highlighted poisoning. A study analysed suicide trends in drug poisonings in Spain between the years 2000 and 2018, showing an annual increase of 7.7% in older adults [37].

Our analysis of monthly trends and the impact of the pandemic did not reveal significant changes at an overall level between 2017 and 2022. Only a small but significant 0.5% increase in the trend of female suicide was noted. However, in a Spanish study between the years 2016 and 2020 also using INE data, the peaks of suicide were found in spring and summer months before 2020. In this year, there was a small change, increasing between late spring and early fall, hypothetically attributed to factors like fear of contagion, isolation, loss, and bereavement as this period was subsequent to the lockdown in Spain [18].

Although overall figures do not show a major change, analyses by age group revealed an increased risk during the pandemic in children and adolescents (0–15 years), as was reported in a 2-year longitudinal study [16] and in men aged 60–64 years. This finding could reflect the combined impact of educational disruption, social isolation, and economic uncertainty. For women, the 23% increased risk in the 45–49 age group suggests possible challenges related to care responsibilities or job insecurity, as has also been noted in other countries [38].

### 4.2. Comparison with Japanese Data

This comparison seeks to identify common patterns and differences influenced by cultural, social, and economic factors that may help to better understand the risk factors for suicide in Spain and Japan. In the first place, one point in common with all countries, and also between Spain and Japan, is the marked difference in suicide rates between men and women; men always predominate in all age groups [39]. Giving an answer to the research questions and starting with the main differences in the frequency of suicides, during the selected period, 2017–2022, suicide rates remained relatively stable in the Spanish population, indicating no significant changes, except the slight increase after the onset of the COVID-19 pandemic in 2020. This suggests that the pandemic may have had a negative impact on the mental health, overall, in the age group of 0–15 years old in males.

The effects of the pandemic on suicide were not uniform, affecting men and women differently across ages, with a greater impact at the extremes of life in men (children/adolescents and older adults) and a modest increase in middle-aged adult women. Overall, the impact of the pandemic on women was less pronounced or not as clear in other age groups different from 45–49 years old.

The significant cultural and social differences that influence suicide rates in Spain and Japan allow an enriched comparison by identifying patterns and variations according to sex, age, methods, and temporality. Comparing this information with the trends in suicides in Japan, an article highlighted the excess suicides post-pandemic, coinciding with the trends in men’s suicide rates aligned before and after the pandemic. In women, the peaking group was younger than that noted in the Spanish population, namely, 20–29 years old. But, during the year 2022, the highest risk changed to older groups (30–69 and above 80 years old) [40]. This mortality was also studied related to unemployment in Japan [41,42,43]. The results showed that during the period 2020–2022, both male and female suicide death rates decreased in individuals in their 20s to 60s, but numbers started to increase in teenagers even before the pandemic. This is in contrast to that noted in males in their 80s and females in their 70s with rates that decreased both before and during the pandemic. Suicided in women in working-age groups increased significantly with the pandemic outbreak [42]. Although, in those studies, a much longer time range was used for the comparison pre-pandemic than post-pandemic.

These sex differences also answer the second research question regarding this aspect. Men had a suicide rate three times higher than women in the Spanish population. A significant difference was the increase of 23% in suicide risk in females in the age group 45–49 years old during the pandemic. In a study that examined suicide mortality in Japan from March to June 2020, they observed unchanging suicide rates at the beginning of the pandemic in both sexes, decreasing from April to May in males and increasing in June in females compared to the values in April and May [44]. Since 2020, the increased in female suicide rates was also evident in the Japanese population until 2021 in all age groups except for the age groups under 20 and over 70 years old. As for men, until 2021, they showed high suicide rates in the age group of 20 to 29 years, but after this year it moved to older age groups (50 to 59 and over 80) [40].

To address the question of whether there are significant differences in the method and time of year for suicides, there were no notable changes between months in suicide rates in the Spanish population associated with the start of the pandemic in both sexes, showing stable patterns over time. Only a slight fluctuation was noted before the pandemic in men, but no drastic changes were noted after the start of the pandemic. However, in the Japanese population, in 2020, in the 3-month period, a notable increase in suicide deaths rates was observed in females between 10 and 40 years old [41,42]. Changes in suicide rates were studied comparing the year COVID-19 pandemic started (2020) with the previous 2 years, finding the months of June and July of 2020 as times of particularly high incidence. However, for women, the largest increase was noted in October [45].

Apart from a slight increase in the most common suicide methods, namely, hanging and drowning, after 2020, there were no significant changes in the methods of suicide among males in Spain. For women, jumping and crashing were the most used methods, although poisoning increased until 2020. Japanese studies that address more recent dates do not emphasize the most commonly used method, but previous studies highlight changes in poisoning in the 1950s and 1960s among youngsters aged 15–29 years, with rates being more stable at older ages [46]. Hanging was noted as the most common method in both sexes based on data from 1999 [47]. Hanging and jumping suicides increase in all ages among both sexes from 1990 until the year 2000, and an increase in the use of other methods, such as overdose or gases, was noted in youngsters between 15–24 and 25–44 years old from 2000 until 2011 [48]. The highest suicide rate by hanging was observed in the 19–20-year-old group based on data obtained between 1979 and 2016 [11].

Based on the multivariate analysis of suicide, in the Spanish population, we could highlight that there was no considerable change in suicide rates over time before the pandemic given baseline time trends of male suicide rates, and a statistically significant effect on male suicide rates was not observed during the pandemic period. The interaction between time and the pandemic reinforced the conclusion that the pandemic had no notable effect on male suicide rates over time. Regarding women, the baseline time trend also showed no significant relationship with suicide. However, during the pandemic period, although the association was not significant, it suggests a slight upward trend in suicide risk. Additionally, the interaction between time and the pandemic period for women was statistically significant, suggesting that the pandemic might have caused a small change (+0.5%) in the suicide trend in women. In the Japanese population, from May to August of 2020, suicide rates decreased in both sexes, but then started to increase, especially for women from September to December of 2021 [49]. One study examined monthly mortality rates in Japanese elderly populations in 2020, showing highest decreases from February to May in both sexes, and an unusual increase was noted in the last quarter of the year [50].

With respect to age-specific patterns, the highest risk was observed in the very old people group, 90–94 years old, in men and the middle-aged group in women; however, this was close to that noted for the transition to the older adult category of 55–59 years old. For men, the lowest risk group was 30–39 years old, still young adults. In general, in both sexes, the suicide risk increases with age, and the lowest risk group in men (0–14 years old) was younger than that noted in women (15–29 years old). In another study that placed special emphasis on the years surrounding the pandemic, the increase in mid-age adult suicides was highlighted, in addition to other sociodemographic factors [19]. The excess in mortality during the year of the pandemic (2020) was also studied in Japan compared to previous years (2011–2019), finding the peaks women in their 20s and 30s–40s [51]. During the first period of pandemic, 2020–2021, excess mortality among Japanese females in all age groups was evident, except at the extremes (under 20 and over 70 age brackets), while more subtle variations were observed in men compared to previous years (2018–2019) [40].

To explain these variations, some authors studied and presented some of the risk factors that were exacerbated in these populations. In Spain, risk factors like social distancing measures, economic impact [52], and symptoms of depression and anxiety derived from loneliness [53] are mentioned. While in Japan, apart from social isolation, the lack of care facilities for older people with needs is cited [51].

The COVID-19 pandemic revealed important lessons regarding the relationship between global health crises and mental health, particularly regarding suicide rates. Although suicide rates did not experience a significant change overall in the Spanish population, the pandemic showed differentiated effects between age groups and by sex. These findings underline the importance of recognizing how public health crises can unequally influence different demographic groups. The comparison between Spain and Japan indicates that suicide methods and mortality peaks varied not only according to the cultural context, but also according to the timing of the pandemic.

### 4.3. Practical Implications

Comparisons of the incidence of suicide rates in different population groups between countries sharing similar economic and social conditions in times of crisis but with different social and cultural environments, like Spain and Japan, can guide health authorities to examine what kind of characteristic interventions targeting specific age or sex population groups should be implemented.

By studying data during the same time periods, differences in these behaviours can provide clues about which protective or risk factors may be favoured by these circumstances due to the peculiarities of each country and can guide the creation of policies to mitigate these influences. In certain times of the year or during specific events, like economic crisis, when the suicide rates can be exacerbated, health services can be strengthened depending on the population’s demands.

This kind of study facilitates cross-cultural exchanges of successful intervention strategies assessing the impact on the incidence rate after crisis intervention services.

### 4.4. Limitations

The main limitation of this study would be the difficulty of comparing suicide data from different countries that may use different collection methods following their own criteria as well as the differences in the collection method from one year to another, as is clearly highlighted in other studies [54]. Regarding the impact of the COVID-19 pandemic, its effects may be influenced by unmeasured contextual factors (e.g., lockdown severity). Our results only reflect descriptive patterns and observed trends and do not intend to establish causal relationships.

When conducting an epidemiological study comparing Western and Eastern countries, one of the key limitations is the methodological challenge of ensuring that the population groups are comparable despite having different demographic profiles (age distribution, sex ratios, and cultural backgrounds), thus affecting the interpretation of data. However, standardization methods and advanced statistical techniques can help in making the populations more comparable.

## 5. Conclusions

In this observational study on suicide in Spain and Japan between 2017 and 2022, significant differences and similarities in suicide rates and associated factors have been revealed in both countries. A common trend in suicide rates based on sex was observed in both countries, but the changes in suicide rates before and after the pandemic were more significant in Japan than in Spain. Spanish women experienced more substantial changes in suicide methods compared to men. In contrast, Japan showed less variations in the predominant method between sexes. Also, older age groups and younger women in Japan were at higher risk, with the suicide risk for men shifting from younger to older ages between 2020 and 2021.

The increase of suicide rates in Japanese during the period 2020–2022 was attributed not only to pandemic-related factors (health problems, unemployment) but also to vulnerability factors already existing before the pandemic [42]. Working long hours has also been a famous cause of suicide in the Japanese population. However, studies with specific populations did not support the idea that reducing working hours would be related to an improvement in suicidal ideation, so other work-related factors could be influential, in addition to the time spent working [55].

Another important part of understanding epidemiological trends in suicide is the analysis of interventions implemented in different countries [56]. Studies comparing the different characteristics of interventions carried out in different countries also provide an overview of the strategies used to prevent suicide and may offer a partial explanation for epidemiological variations, since different intervention strategies could influence suicide rates and the most commonly used suicide methods in each country.

As previously mentioned, Japan has implemented advanced suicide prevention programs. Studying specific trends in both countries offer the opportunity to adapt these effective strategies to the Spanish context. Future interventions should target specific vulnerable groups, such as young women and older men in Japan, as well as younger populations in Spain. Geographically, efforts could focus on regions with higher socio-economic disparities or limited access to mental health resources to minimize associated risk factors. A good strategy to reach more complex populations, both due to their geographic location and age, is the gatekeeper-training programs in Japan [57]. In addition, the use of virtual means (video calls, telephones, social media messages) and measures like those used in Japan [58] can help applying guidelines on media coverage to manage the spread of information online and the responses to it, especially in response to youth-related issues.

Mental health professionals can use this information to better identify risk factors in their patients and tailor their treatment approaches based on specific characteristics of sex, age, methods, and months during which the person may need extra support or supervision. Different age groups may face different social, economic, and personal pressures that affect suicide risk. Comparing these data can help identify why certain groups are more vulnerable in each country and the reasons.

### Future Research

The disparity observed between Spain and Japan in suicide rates between men and women in different age groups highlights the need for suicide prevention interventions and policies that consider sex differences. Establishing international collaborations and sharing data and methodologies can facilitate more accurate and comparable analyses to explore variables of interest in common problems such as suicidal behaviour.

Compared to the Japanese population, the relative stability of rates over time in the Spanish population despite a slight recent increase underscores the importance of continuing to monitor and analyse suicide rates to detect trends and changes that may require immediate action.

Conducting long-term longitudinal studies to follow the changes observed for suicide rates as well as comparative analyses of suicide prevention policies and interventions can offer together a more complete and comprehensive understanding of suicide and enhance the effectiveness of prevention efforts among those countries that are compared, in this case, Spain and Japan. While this study does not constitute a direct comparative analysis, descriptive insights from both countries highlight potential areas for further investigation and intervention.

## Figures and Tables

**Figure 1 healthcare-13-00451-f001:**
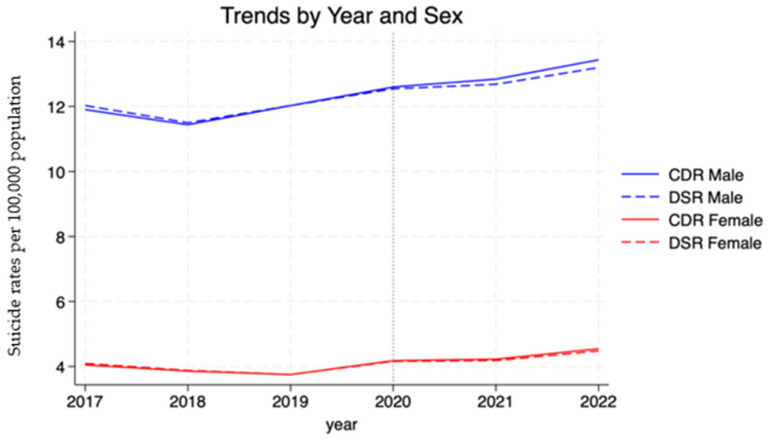
Crude and directly standardized suicide rates.

**Figure 2 healthcare-13-00451-f002:**
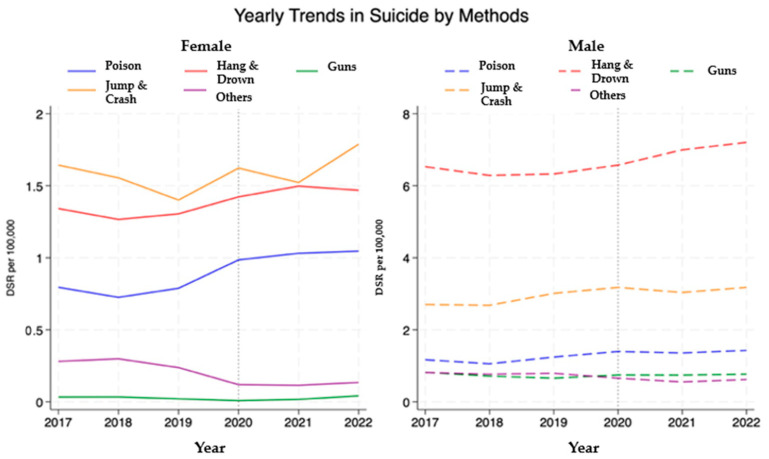
Trends in suicide mortality by method and sex.

**Figure 3 healthcare-13-00451-f003:**
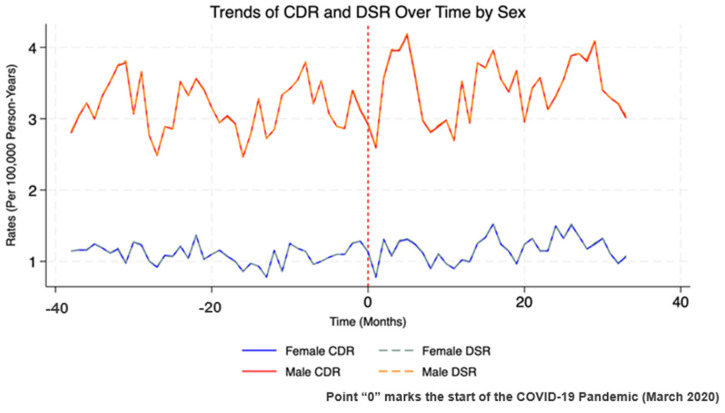
Time trends in suicide crude and directly standardized mortality rates.

**Figure 4 healthcare-13-00451-f004:**
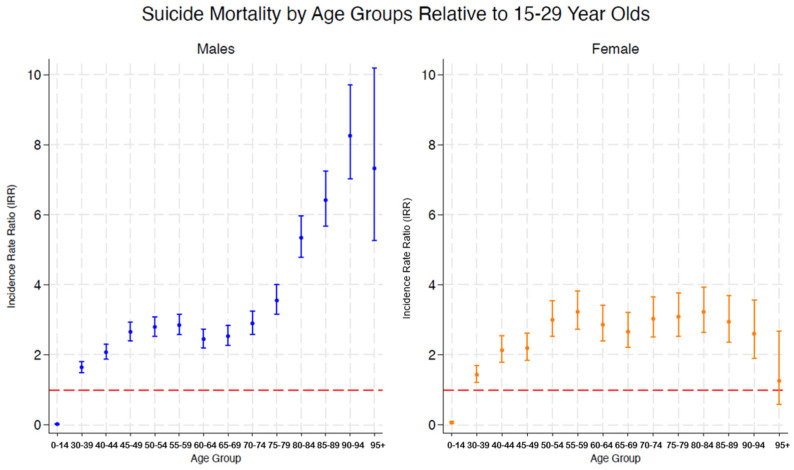
Risk of suicide by age group relative to age 15 (15–29 years).

**Figure 5 healthcare-13-00451-f005:**
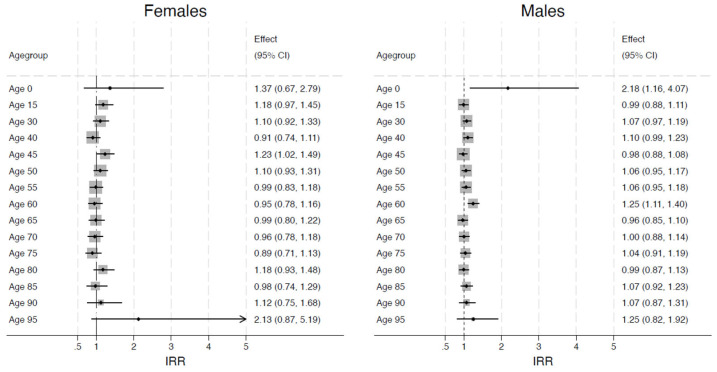
Effect of the pandemic on suicide rates by age group.

**Table 1 healthcare-13-00451-t001:** Multivariate analysis of suicide mortality in males.

Variable	IRR	*p*-Value	Lower CI	Upper CI
Time	0.9998	0.819	0.9978	1.0017
Pandemic	0.9892	0.859	0.8779	1.1148
Pandemic#Time	1.0022	0.144	0.9993	1.0051
Age Group	-	-	-	-
Age 15 (Reference)	1	Reference	Reference	Reference
Age 0	0.0224	0	0.0134	0.0373
Age 30	1.6442	0	1.4923	1.8115
Age 40	2.0729	0	1.8713	2.2962
Age 45	2.6546	0	2.4063	2.9285
Age 50	2.7958	0	2.5335	3.0853
Age 55	2.8481	0	2.575	3.1502
Age 60	2.4498	0	2.1962	2.7327
Age 65	2.5323	0	2.2622	2.8345
Age 70	2.8969	0	2.5866	3.2445
Age 75	3.5525	0	3.1565	3.9981
Age 80	5.3424	0	4.7791	5.9721
Age 85	6.4176	0	5.6787	7.2526
Age 90	8.2556	0	7.0228	9.7049
Age 95	7.3236	0	5.2608	10.1953
Agegroup#Pandemic	-	-	-	-
Age 0#1	2.1995	0.015	1.1685	4.1404
Age 30#1	1.0844	0.255	0.9431	1.247
Age 40#1	1.116	0.141	0.9643	1.2914
Age 45#1	0.987	0.856	0.8573	1.1365
Age 50#1	1.0668	0.367	0.927	1.2276
Age 55#1	1.069	0.361	0.9264	1.2334
Age 60#1	1.2587	0.003	1.082	1.4642
Age 65#1	0.9754	0.764	0.8293	1.1472
Age 70#1	1.0117	0.888	0.8603	1.1898
Age 75#1	1.0506	0.559	0.8901	1.2402
Age 80#1	1.0038	0.964	0.8532	1.181
Age 85#1	1.0791	0.388	0.9079	1.2825
Age 90#1	1.0815	0.494	0.8642	1.3534
Age 95#1	1.2643	0.293	0.8169	1.9566

**Table 2 healthcare-13-00451-t002:** Multivariate analysis of suicide mortality in females.

Variable	IRR	Std. Err.	*p*-Value	Lower CI	Upper CI
Time	0.9982	0.0017	0.281	0.995	1.0015
Pandemic (1)	1.1827	0.1212	0.102	0.9675	1.4458
Pandemic#Time (1)	1.0057	0.0025	0.024	1.0008	1.0107
Age Group	-	-	-	-	-
Age 15 (Reference)	1	Reference	Reference	Reference	Reference
Age 0	0.0647	0.0178	0	0.0377	0.1111
Age 30	1.4309	0.1263	0	1.2036	1.7011
Age 40	2.131	0.1925	0	1.7852	2.5438
Age 45	2.1929	0.199	0	1.8356	2.6197
Age 50	2.998	0.2566	0	2.535	3.5456
Age 55	3.2276	0.2775	0	2.727	3.8201
Age 60	2.8601	0.2609	0	2.3919	3.42
Age 65	2.6595	0.2539	0	2.2056	3.2068
Age 70	3.0297	0.2866	0	2.517	3.6469
Age 75	3.0891	0.3119	0	2.5346	3.765
Age 80	3.2259	0.3262	0	2.6459	3.933
Age 85	2.9438	0.3375	0	2.3515	3.6854
Age 90	2.6024	0.4174	0	1.9004	3.5635
Age 95	1.2578	0.4827	0.55	0.5928	2.6687
Agegroup#Pandemic	-	-	-	-	-
Age 0#1	1.1547	0.43	0.699	0.5566	2.3958
Age 30#1	0.9342	0.1156	0.582	0.7331	1.1906
Age 40#1	0.7662	0.0989	0.039	0.595	0.9867
Age 45#1	1.0426	0.1294	0.737	0.8175	1.3297
Age 50#1	0.9339	0.1107	0.564	0.7404	1.1781
Age 55#1	0.834	0.1001	0.13	0.6592	1.0551
Age 60#1	0.8037	0.1026	0.087	0.6259	1.0322
Age 65#1	0.8347	0.112	0.178	0.6417	1.0859
Age 70#1	0.8109	0.1082	0.116	0.6243	1.0532
Age 75#1	0.7553	0.1076	0.049	0.5713	0.9985
Age 80#1	0.9945	0.1409	0.969	0.7533	1.3128
Age 85#1	0.8246	0.1339	0.235	0.5998	1.1336
Age 90#1	0.9489	0.2083	0.811	0.6171	1.4592
Age 95#1	1.7982	0.8314	0.204	0.7266	4.4503
Constant	1.63 × 10^−6^	1.21 × 10^−7^	0	1.41 × 10^−6^	1.89 × 10^−6^

## Data Availability

The following link provides access to the OSF platform where the protocol that was followed for the analysis of this study is registered: https://osf.io/kzeb5/?view_only=9879d513ab374dbea9cad2d13dc58f12 (accessed on 29 January 2025).

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
