# Peer review of "Epidemiological Analysis of Suicidal Behaviour in Spain from 2017 to 2022 and Comparative Perspectives with Japan: A Retrospective Observational Study"

_healthcare, 2025, doi:10.3390/healthcare13050451_

Round 1
Reviewer 1 Report
Comments and Suggestions for Authors
Suicide is a significant public health globally issue, and patterns of suicide may have changed with the onset of the COVID-19 pandemic, and through epidemiological comparative studies we can identify patterns and disparities. AUTHORS’ study is a comparative epidemiological analysis of suicide in Japan and Spain during the period from 2017 to 2022 aiming to explore the variability of the distribution of suicides in both countries.
THEIR study uses vital registration data from Spain to estimate trends in suicide mortality over this period, and to calculate changes in trends as well as methods (based on ICD) to compare with those in Japan.
THEIR results show:
1) In Spain, no significant changes in suicide rates were found before and after the pandemic period, but there were during the beginning of the pandemic (monthly) in the youngest, with more substantial changes in the method in women than in men.
2) in Japan a notable variation was found following the pandemic. Older age groups among the Japanese population and younger women were at higher suicide risk, while in men, the suicide risk shifted from younger to older ages between 2020 and 2021.
3) As a common point, a similar trend was observed in both countries on sex with different influence on women.
AUTHORS conclude that: (A) their study offers a detailed insight into the patterns and circumstances of suicide in these countries, offering a basis for future interventions and preventive measures. (B) These comparisons emphasize the critical nature of suicide and underline the necessity for enhanced prevention efforts.
The Covid-19 pandemic has changed our lives and has also had a significant impact on the risk of suicide.
This study addresses the topic of the relationship of suicides in two different nations in terms of culture and geography (Spain and Japan) and this gives a first added value.
The study is very interesting and useful (also in perspective)
I have the following comments for the authors:
1) Enhance in all sections the choice (which I share) to focus on Spain and Japan
2) Try to give a better structure to the introduction, go for clear passages (suicide trend in the world, impact of the pandemic, focus on the two nations).
3) Good search questions ... but follow up by inserting a nice robust purpose.
4) Section 2 should be developed more broadly by avoiding sections of a single sentence (see section 2.1 erroneously numbered with 1.1). This in my opinion to enhance the entire project a little better. There are variables, statistics, data sources…but a clear and robust design development section is missing.
5) The results are interesting but should be grouped into themes. Also, some figures should be improved both in resolution and with data-labels.
6) Nice discussion. I suggest organizing some considerations in a paragraph like “lesson learned from—covid-19..”
Minor points: The references must be reacalled by [] not by (). Check the MDPI standard in the text and in the references
Reviewer 2 Report
Comments and Suggestions for Authors
This article offers a comparative analysis of suicide trends in Spain and Japan from 2017 to 2022, focusing on the impact of the COVID-19 pandemic. The authors aim to identify patterns and disparities in suicide rates across the two countries using vital registration data. While the findings are presented clearly, several improvements can be made to enhance the clarity, precision, and depth of the abstract.
The structure of the abstract could be improved to make the information more organized and easier to follow. Dividing it more clearly into sections such as background, methods, results, and conclusions would help improve readability. In addition, the abstract mentions that there were "substantial changes in methods" in Spain, but does not specify what these changes were, especially in relation to gender differences. It would be helpful to provide concrete examples of the methods that saw an increase and how these changes differed between men and women.
Regarding Japan, the abstract mentions a shift in suicide risk from younger to older age groups in men between 2020 and 2021. The term "shifted" is vague and could benefit from a more detailed explanation. The authors could expand on the factors that contributed to this shift, such as socio-economic factors, mental health, or other variables. Additionally, the statement that younger women were at higher suicide risk in Japan could be further explored, particularly in the context of the pandemic's specific impact on this group.
The abstract also touches on the influence of the pandemic but does not consider other potential environmental or policy factors that could have played a role in the observed trends. Including a brief mention of these factors, such as government interventions or economic support, would provide a more comprehensive understanding of the shifts in suicide rates.
Another point to address is the methodology used for comparison between Spain and Japan. While the abstract mentions vital registration data and ICD classifications, it does not elaborate on how the data were analyzed or whether additional factors like regional differences or demographic characteristics were considered. Clarifying the methodology would strengthen the study’s scientific rigor.
Finally, the conclusion emphasizes the need for future interventions and prevention efforts, but this point could be developed further. Instead of merely stating the need for action, the authors could specify which groups (e.g., age, gender) or geographical areas require targeted interventions. This would make the conclusion more actionable and relevant to policy and practice.
In summary, this study offers valuable insights into suicide trends in two distinct countries. However, a few revisions could improve the clarity and comprehensiveness of the abstract. These include reorganizing the content for better flow, providing more specific details on trends and shifts, addressing the role of environmental and policy factors, and elaborating on the methodology and conclusions. With these adjustments, the abstract would become more precise and compelling, offering a stronger basis for future interventions and preventive measures.
Reviewer 3 Report
Comments and Suggestions for Authors
This study appears promising, as the authors describe in the title and abstract an analysis of suicide data associated with the COVID-19 pandemic in Japan and Spain. However, this research presents significant issues.
To my mind, the results are merely an analysis of broader statistics and do not provide novel information for the scientific community. In other words, there is no reporting of important data on COVID-19 pandemic-related suicides, as the title of the manuscript promises. Instead, the authors analyze suicide mortality from 2017 to 2021, focusing on differences between men and women.
Additionally, the authors mention an analysis of data from Spain and Japan, but they only analyze data from Spain and then compare it with Japanese data in the discussion section. Therefore, this is not a genuine comparative study.
Moreover, the manuscript does not meet the quality standards required for publication, as there is no reference to adherence to STROBE guidelines, ethical committee approval, or similar protocols. Many sections need to be reviewed, expanded, and rewritten for greater clarity.
For all these reasons, the reviewer’s decision is to reject the manuscript and request that the authors improve it before resubmission for reevaluation.
Below, more detailed points for improvement are provided.
ABSTRACT:
The results section should be rewritten. The suicide data comparison between the two countries is somewhat confusing, particularly for Japan.
INTRODUCTION:
This reviewer recommends reorganizing the information in this section. The authors begin by discussing global figures, then the importance of epidemiological studies, and subsequently return to specific country data. In the reviewer’s opinion, this back-and-forth between topics is disorganized. These shifts in focus occur again later, where the authors discuss sociodemographic factors related to suicide and abruptly switch to the methods used for suicide.
The reviewer also advises the authors to be more cautious when writing sections such as the introduction, methods, and abstract, as they tend to write as if expressing personal opinions. For instance, in the phrase, “An event that not surprisingly increased the number of suicides in Spain was the pandemic,” they imply that the increase in suicides during COVID-19 is unsurprising. How do the authors reach this conclusion? In these sections, information must be presented objectively, reserving opinions and conclusions for the discussion section.
Additionally, this reviewer suggests combining shorter paragraphs into longer ones of at least 7–8 lines.
The authors should also review the journal’s citation guidelines. For example, citing the article by De la Torre-Luque et al. (2024) (14) is inconsistent. Either the name and year format or the reference number should be used.
Once again, the authors jump between ideas, mixing factors that influence suicide. They first discuss gender, then age, and then return to gender. This makes it difficult for the reader to follow the logical flow of the introduction.
The authors need to structure their objectives better in alignment with the introduction. Objectives should be the last part of the introduction. The information provided after the objectives, which discusses how this study’s data could help develop better strategies to reduce suicide rates, should be placed in the conclusion or discussion sections.
Section 1.1 should be integrated into the objectives; it should not exist as a separate section. The reviewer suggests the authors build their objectives and hypothesis more solidly. As a reader, it is difficult to understand the study’s purpose, particularly given the erratic flow of ideas in the introduction. It is unclear whether the authors aim to examine how COVID-19 increased suicide rates or to simply provide a snapshot of suicides during the lockdown or the pandemic. It is also important to note that lockdown periods were not the same in Spain and Japan.
MATERIALS AND METHODS:
This section lacks a subsection explaining the study design, participant data, and other essential aspects of an epidemiological study. Without this information, the manuscript cannot be published in any quality journal.
Please provide information about adherence to STROBE guidelines, which are fundamental and must be followed rigorously in epidemiological research.
Additionally, the authors mention analyzing variables such as gender but do not include the analysis of other relevant sociodemographic variables. The reviewer questions how it is possible for the authors to emphasize factors such as socioeconomic status in the introduction but not include them as part of the study variables.
The authors do not indicate the level of statistical significance or other relevant statistical metrics.
RESULTS:
The authors do not provide a description of the sample.
Once again, the results are presented in a confusing manner, similar to the introduction, with a mix of ideas that hinder the reader’s understanding of the data.
The reviewer questions why the authors place so much emphasis on differences in suicide rates between Spain and Japan, but then present the data aggregated without addressing the particularities of each country.
DISCUSSION:
The authors compare their data with Japanese statistics. However, this reviewer does not understand the purpose of this comparison. If the title indicates that suicide in Spain and Japan is analyzed, why do the authors only analyze Spanish data and then compare it to Japanese data in the discussion? Shouldn't the analysis of both countries’ data come first, followed by a discussion of these findings in comparison to other populations?
The conclusions should be adjusted to reflect the reality of the study: it is not a true comparative study.
The section on future research should be eliminated, and the information distributed across the discussion or limitations section.
